# Fanconi Anemia: Examining Guidelines for Testing All Patients with Hand Anomalies Using a Machine Learning Approach

**DOI:** 10.3390/children9010085

**Published:** 2022-01-07

**Authors:** Christoph Wallner, Jane Hurst, Björn Behr, Mohammad Abu Tareq Rony, Anthony Barabás, Gill Smith

**Affiliations:** 1Department of Plastic Surgery, Great Ormond Street Hospital, London WC1N 3JH, UK; gos-tr.clinicalgenetics@nhs.net (J.H.); tonybarabas@googlemail.com (A.B.); gill.smith@gosh.nhs.uk (G.S.); 2Department of Plastic and Hand Surgery, Burn Center Sarcoma Center, BG University Hospital Bergmannsheil Bochum, Ruhr University Bochum, 44789 Bochum, Germany; bjorn.behr@rub.de; 3Department of Statistics, Noakhali Science and Technology University, Noakhali 3814, Bangladesh; abutareqrony@gmail.com

**Keywords:** Fanconi, hand surgery, pediatric malformation

## Abstract

Background: This study investigated the questionable necessity of genetic testing for Fanconi anemia in children with hand anomalies. The current UK guidelines suggest that every child with radial ray dysplasia or a thumb anomaly should undergo further cost intensive investigation for Fanconi anemia. In this study we reviewed the numbers of patients and referral patterns, as well as the financial and service provision implications UK guidelines provide. Methods: Over three years, every patient with thumb or radial ray anomaly referred to our service was tested for Fanconi Anemia. CART Analysis and machine learning techniques using Waikato Environment for Knowledge Analysis were applied to evaluate single clinical features predicting Fanconi anemia. Results: Youden Index and Predictive Summary Index (PSI) scores suggested no clinical significance of hand anomalies associated with Fanconi anemia. CART Analysis and attribute evaluation with Waikato Environment for Knowledge Analysis (WEKA) showed no single feature predictive for Fanconi anemia. Furthermore, none of the positive Fanconi anemia patients in this study had an isolated upper limb anomaly without presenting other features of Fanconi anemia. Conclusion: As a conclusion, this study does not support Fanconi anemia testing for isolated hand abnormalities in the absence of other features associated with this blood disease.

## 1. Introduction

Fanconi anemia (FA) is a very rare, inheritable, primarily autosomal recessive anemia that leads to bone marrow failure. The incidence of FA is estimated to be around 1 in 136,000 births [1].

Although considered primarily a blood disease, FA may affect different organs of the body. Data from 1865 case reports in the literature suggest that the most frequent characteristic birth defects in FA (in descending frequency from approximately 50 to 20 percent) include skin hyperpigmentation and café au lait spots; short stature; abnormal thumbs and radii; microcephaly; and eye, kidney and ear abnormalities [2]. However, reports are biased as cases published in the literature tend to focus on more unusual or sensational findings.

The UK FA guidelines state that 71% of FA patients have skeletal anomalies, which include radial ray dysplasia (RRD), but without any emphasis on specific anatomical areas [3]. US guidelines indicate that approximately 50% of FA children have skeletal anomalies, and that 70% of these are upper extremity anomalies, most commonly affecting the thumb and radial border of the forearm [2]. Accordingly, 35% of FA patients would be expected to have an upper limb anomaly. In addition, US guidelines also state that syndactyly or other abnormalities of the toes are also a feature of FA [2]. However, none of these statements are referenced.

Both the UK and US guidelines suggest that every child with RRD or thumb anomaly, in which no alternative causative genetic disorder has been identified, should undergo further investigation to determine whether they have FA [2,3,4]. Children should be referred to a clinical geneticist or directly for FA testing using a chromosome breakage test (CBT). This study reviews the numbers of patients and referral patterns, as well as the financial and service provision implications these guidelines have on our pediatric service.

## 2. Materials and Methods

### 2.1. Patients

Permission was obtained from the ethics and audit committees to undertake a retrospective study of all patients who attended the congenital hand anomalies service over a period of three years (1 January 2010 to 31 December 2012). The unit is one of the largest children’s hospitals in Great Britain. Every new patient presenting with a thumb or radial ray anomaly was identified during this period, and the number and pattern of referrals to investigate a possible diagnosis of FA were examined.

The results of peripheral blood chromosome breakage test (PB-CBT) for FA sent from all departments at our service over the same time were also obtained and analyzed. The clinical features triggering referral for PB-CBT testing were examined for each patient tested, and the features of those patients testing positive for FA were analyzed.

### 2.2. CART Analysis and Feature Importance

CART analysis was used for binary recursive partitioning in Python (version: 3.8.3). To evaluate feature importance (correlation attribute evaluation), the Waikato Environment for Knowledge Analysis (WEKA) was implemented with the correlation attribute evaluation technique, which requires the use of a Ranker search method (cut-off = 0.1, threshold = −1.79, num to select = −1) [5].

### 2.3. Statistical Analysis

Data management was performed using GraphPad PRISM (version: 8.3.0; Graphpad Software, Inc., La Jolla, CA, USA).

## 3. Results

### 3.1. Patient Demographics

Over the three years study period, 195 new patients with a diagnosis of a thumb or radial ray abnormality were referred to the congenital hand service, of which only one-quarter (*n* = 49) had a genetic disorder to explain the anomaly. Of the 146 patients with no known genetic diagnosis, only 9 (6%) were referred directly for PB-CBT testing and a further quarter (*n* = 48) were referred on to the genetics department at the discretion of the senior author. The genetics department subsequently referred 27 patients with hand anomalies on for PB-CBT testing (thumb duplication: *n* = 9; thumb hypoplasia: *n* = 11; RRD: *n* = 6; little finger brachydactyly: *n* = 1); however, except four patients, all other patients had other features than hand anomalies associated with FA (Cranial abnormalities: *n* = 10; short stature: *n* = 3; other skeletal abnormalities: *n* = 8; abnormal skin pigmentation: *n* = 4; visceral organ abnormalities: *n* = 8). One of these 27 patients tested positive for FA. This patient had RRD, a tracheaoesophgeal fistula, renal agenesis, short stature, and a cleft palate. One patient referred directly for PB-CBT also tested positive for FA. This patient had a thumb duplication associated with cardiac anomalies. In addition, none of those patients testing negative for FA have been referred for a second PB-CBT test, contrary to both UK and US guidelines.

### 3.2. Referral Patterns

Over this period, 169 patients (median age: 6 ± 7.5 years) from the different departments at our hospital were tested for FA by PB-CBT. Most referrals were from the genetics (*n* = 69) and hematology departments (*n* = 56), followed by hand surgery (*n* = 9) and dermatology (*n* = 8). Nine referrals came from other departments, and in 18 patients the source of referral was unknown. The features that led to testing in each case were further examined.

### 3.3. Clinical Findings in the Patient Collective

Of the 169 patients referred to our hospital for PB-CBT, 42 (25%) had abnormalities of the upper limbs. These anomalies included thumb duplications (*n* = 13), thumb hypoplasia (*n* = 17), and radial ray dysplasia (*n* = 11). Features that led to genetic testing were skeletal anomalies (excluding the upper limbs) (*n* = 16), craniofacial anomalies (*n* = 49), small stature (*n* = 20), aberrant skin pigmentation (*n* = 39), visceral anomalies (*n* = 20), and anemia (*n* = 40).

### 3.4. Clinical Features in Patients Tested Positive for Fanconi Anemia

Thirteen (median age: 6 ± 7.5 years) of the 169 (8%) patients referred for PB-CBT tested positive for FA (see Figure 1). Three of these FA positive patients had upper extremity abnormalities, including one patient with radial ray dysplasia on both hands and two patients with duplication of the thumb. However, all three patients presented visceral, skeletal, or hematological abnormalities, which are found to have a higher association with FA than isolated upper extremity anomalies (hand abnormality—23%; microcephaly—31%; small stature—31%; consanguinity—39%; cardiac/renal abnormalities—46%; abnormal skin pigmentation—46%; blood dyscrasia—62%). No patient with an isolated thumb anomaly tested positive for FA. Six FA patients were positive for familial consanguinity, three of which had been referred for testing due to a positive proband sibling.

### 3.5. Hand Anomaly Is Not a Predicting Feature for Fanconi Anemia According to Youden Index and Predictive Summary (PSI)

Taken together, our collective results showed no significant correlation between isolated hand related abnormalities and positive Fanconi test results (*p* = 0.522, Chi-square test) (see Figure 2).

Next, we wanted to test the assumption that a hand anomaly can be used as a screening symptom for FA. For that assumption, specificity, precision, accuracy, recall, Youden Index, and Predictive Summary Index (PSI) scores were calculated [6,7]. Negative Youden Index and PSI scores significantly demonstrate that there is no reasonable correlation of a hand anomaly with FA (see Figure 2).

### 3.6. Attribute Evaluation and CART Analysis Showed no Significant Clinical Features for FA

In a next step, we used machine learning analyses to find meaningful relationships with a positive FA testing. While thumb hypoplasia, thumb duplication, and radial dysplasia were highly associated with other hand anomalies and abnormal pigmentation was associated with cranial abnormalities, there was no single feature associated with a positive FA testing in the feature correlation matrix (see Figure 3A).

In order to evaluate the importance of different features, a decision tree regressor (CART) was implemented to calculate feature importance scores. Date of birth and referring specialty were the highest rated predictors for the diagnosis of FA (see Figure 3B).

The attribute evaluation with WEKA showed there was no attribute correlated with FA positive testing (see Figure 3C).

## 4. Discussion

At our center, a genetics consult was requested for all patients with RRD and any patient with an upper limb anomaly with other features suggestive of an underlying genetic diagnosis, even prior to implementation of the current guidelines. However, patients with an isolated thumb anomaly and no other features of FA (or an alternative genetic condition) were not routinely referred to the genetics department or for FA testing. The guidelines challenged this policy. Our approach is supported by this study, which revealed that no patient with an isolated thumb anomaly tested positive for FA. More importantly, none of the 110 patients who were not referred for PB-CBT testing have gone on to develop FA. The average follow-up period is presently only three years. Patients with FA would be expected to have been diagnosed within this time, but without longer follow-up it is not possible to be certain if any patients may yet develop FA.

This study supports the assumption that no statistically significant correlation between FA and isolated hand-related abnormalities exists. A review of the 13 patients testing positive for FA revealed that abnormalities of the upper limb were found to have a weaker association with FA than visceral (renal/cardiac), skeletal (small stature/microcephaly), dermatological (aberrant skin pigmentation or café au lait spots), or hematological anomalies. No patient had an isolated upper extremity anomaly. Three FA positive patients had upper limb anomalies, but these were in association with other features of FA. This supports the argument that isolated upper limb anomalies do not require FA testing. Here, 23% (3/13) of the patients testing positive for FA had an upper limb anomaly, which was less than the 35% expected according to the current guidelines; however, it must be acknowledged that the numbers of positive patients in the study are small.

In this study, the use of supervised machine learning was able to validate the results of the PSI and Youden Index in a multidimensional way. The classification and regression as sub-areas of supervised machine learning enable connections between features to be established more comprehensibly than with previous methods. Even with small amounts of data, as in this study with nominal or binary data, the application of machine learning can be expedient. Therefore, a set of information as given in this study is most suitable to be evaluated through a machine learning algorithm. However, as with many statistical methods, a larger set of data would enhance the quality of the results. Regarding rare diseases such as FA, even as a specialized metropolitan hospital it is difficult to generate a larger data set. The question of whether a hand anomaly is a predictive variable for FA was answerable with machine learning methods in three different ways. Machine learning is becoming an indispensable part of biomedicine, especially regarding complex problems such as genome analysis and image processing [8]. In this context, supervised and unsupervised machine learning are even more appropriate than in our study. The analysis of complex data sets especially benefits from machine learning methods. We were not able to identify any disadvantages compared to conventional statistical analysis in the described methodical approach. One obstacle in implementing machine learning in studies is the required knowledge of programming languages.

As shown in our cohort, more patients present with other features than hand anomalies, including skin pigmentation, visceral abnormalities, and blood dyscrasia. This observation is consistent with the literature. In descending frequency from 50% to 20%, skin abnormalities, short stature, abnormal head, abnormal eyes, abnormal kidneys, and hand or forearm anomalies are observed [4,9]. At least 25% of known FA patients have few or none of these findings. A study on an Italian FA population demonstrated that patients displaying a mild phenotype (43%) outnumbered patients with moderate or severe phenotypes [10]. Schneider et al. provided a more detailed breakdown of the incidence of somatic abnormalities in Fanconi patients. Male and female infertility affect 100% of patients [11]. The authors also describe a short stature, skin abnormalities, or skeletal findings with an incidence range of 60–70%. Hand and forearm findings were subsumed in the study by Schneider under general skeletal abnormalities, which also included hip dysplasia, vertebral sclerosis, and rib deformities [11]. The severity of the phenotypic expression also depends on the mutation. While null mutations lead to a more severe phenotype, mutations in *FANCA* and *FANCC* are associated with lower rates of anomalies [12,13]. One of the most common clinic finding is head and neck squamous cell carcinoma [14,15,16]. Based on our results, neither statistical nor machine learning methods showed a single feature predictive of FA. It is rather important to detect a combination of clinical appearances that indicate the necessity for FA testing other than hand anomalies.

The cost of referring a patient to a clinical geneticist at our service is £500 (US $767). Ninety-eight new patients with thumb abnormalities were not referred for genetics consult. Adherence to the guidelines would have cost the hand department an additional £49,000 over three years (US $75,200) if a genetics referral had been requested for all patients. In comparison, a PB-CBT costs £384 (US $590) and would have cost the department an additional £37,632 (US $57,750). However, UK guidelines recommend that a second PB-CBT needs be performed in all patients initially testing negative for FA. The cost of the PB-CBT is, therefore, doubled, as most patients test negative for FA and would need the test repeated. The rational for repeated testing is to detect the 10–15% of FA patients who are somatic mosaics [3]. In patients with somatic mosaicism, a mutation occurs in an early hematopoietic stem or progenitor cell that leads to correction of the FA chromosome breakage phenotype. This results in the peripheral blood cells appearing to be normal in the PB-CBT when the patient in fact has underlying FA.

The skin biopsy fibroblast chromosome breakage test (SBF-CBT) has a nearly 100% sensitivity and specificity in diagnosing FA [3] at a cost of £500 (US$767). Patients older than 5 years are recommended to have a SBF-CBT instead of a PB-CBT, as the guidelines suggest it is more appropriate to take a skin biopsy in older patients. However, in this study, the majority (92%) of patients with RRD or a thumb anomaly were offered surgical correction under general anesthesia. This provides the ideal opportunity to take a skin biopsy, particularly as a portion of skin may be excised as part of the surgical procedure. This can be easily arranged through good communication between the hand surgeon and geneticist. Economically, a skin biopsy (costs £500 (US$767)) is a better investment than two PB-CBT’s, which may still miss cases of mosaicism (costs £768 (US$1178).

For financial reasons, the hand department has decided to stop direct referral of patients for PB-CBT testing. All patients are instead referred to the genetics department for consideration of FA testing. The geneticists only performed a PB-CBT test on 56% of the patients referred to them from the hand department, having screened out those patients for whom FA testing was felt to be clinically unnecessary. These patients were included in the follow-up period, and none have gone on to develop FA.

## 5. Conclusions

The current FA guidelines have tremendous financial and clinical implications for patients anomalies of the thumb or radial ray [2,3]. There is additionally the psychological cost of causing undue concern for the patient and their family by strictly adhering to the guidelines in all cases of a thumb anomaly or RRD.

None of the thirteen FA positive tested patients in this study had an isolated upper extremity abnormality without presenting other features associated with FA. In addition, in this cohort no patient to date with an isolated thumb anomaly has developed FA. This study delivers striking arguments against the FA testing for isolated thumb abnormalities in the absence of other FA associated features. In cases where FA is suspected, referral to a clinical geneticist is advocated rather than direct referral for FA testing.

Referral is at the discretion of the hand surgeon and it is important that they are aware of and look for features associated with FA. In our practice, this means looking for café au lait spots or unusual skin pigmentation, plotting height and head circumference on age-appropriate centile charts (normally brought in with the parents in the UK “red book”), reviewing blood tests, and less often requesting an ultrasound or cross-sectional study to exclude visceral abnormalities (often required anyway as part of their pre-surgical work-up). Enquiries into possibility parental consanguinity should be made when appropriate.

When a decision is made to test a child for FA, and if that child is to undergo a surgical procedure, obtaining a skin biopsy for SBF-CBT is recommended due to its sensitivity and cost-effectiveness rather than a PB-CBT.

The congenital hand service at our hospital, in conjunction with the genetics department, has devised an algorithm based on our results that is believed to reduce the unnecessary testing of all patients with a congenital thumb anomaly for FA (see Figure 4). The small sample size and the relatively short median follow-up period of this study are recognized. In addition, the median age for developing hematological abnormalities from FA is around age 7 [9]. The hand department monitors most patients on a yearly outpatient basis until skeletal maturity and will continue to evaluate all patients with thumb anomalies who were not sent for FA testing in recognition of this. A larger multi-centered study has been started to increase the power of this study and to further evaluate the association and frequency of congenital anomalies with FA.

## Figures and Tables

**Figure 1 children-09-00085-f001:**
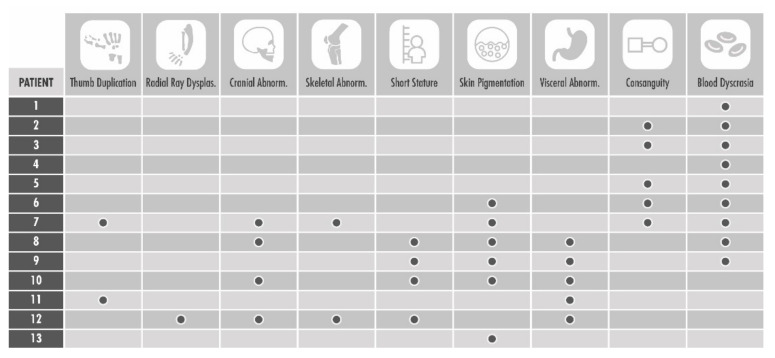
Features of the 13 patients tested positive for Fanconi anemia.

**Figure 2 children-09-00085-f002:**
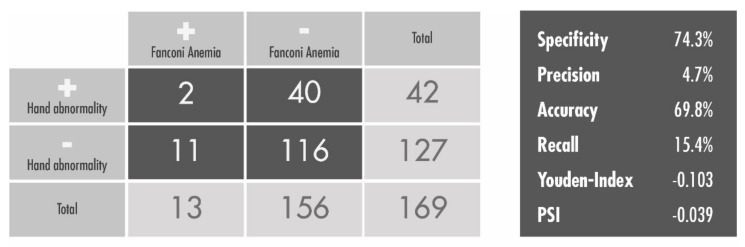
Correlations of hand abnormalities and Fanconi anemia. Chi-square test used to find correlations between isolated hand abnormalities (negative over positive) and Fanconi anemia (negative over positive) in our cohort. The two-tailed *p* value equals 0.522. Specificity, precision, accuracy, recall, Youden Index, and Predictive Summary Index (PSI) scores as test evaluation criteria are shown on the right.

**Figure 3 children-09-00085-f003:**
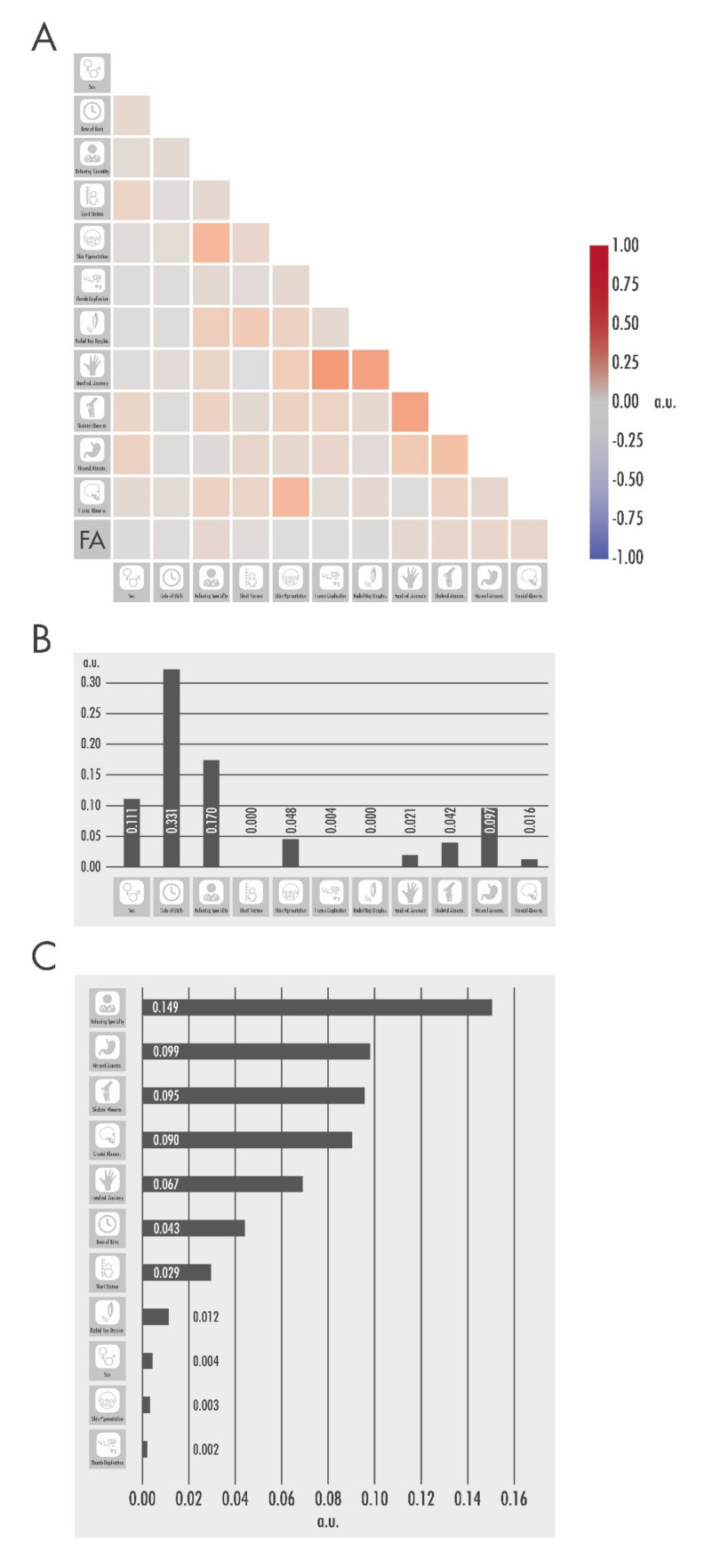
Machine learning approaches used to detect correlations. (**A**) The heat map of the feature correlation matrix unveiled a correlation between hand anomalies, thumb hypoplasia, and radial dysplasia, as well as abnormal pigmentation and cranial abnormality, but no clinical feature was predictive for positive testing of FA. (**B**) CART analysis showed date of birth (score: 0.33), referring specialty (score: 0.17), and cranial abnormality (score: 0.16) as the most important attributes for positive FA testing. (**C**) Attribute evaluation in WEKA with a cut-off at 0.1 showed that no feature predicted a positive FA test result.

**Figure 4 children-09-00085-f004:**
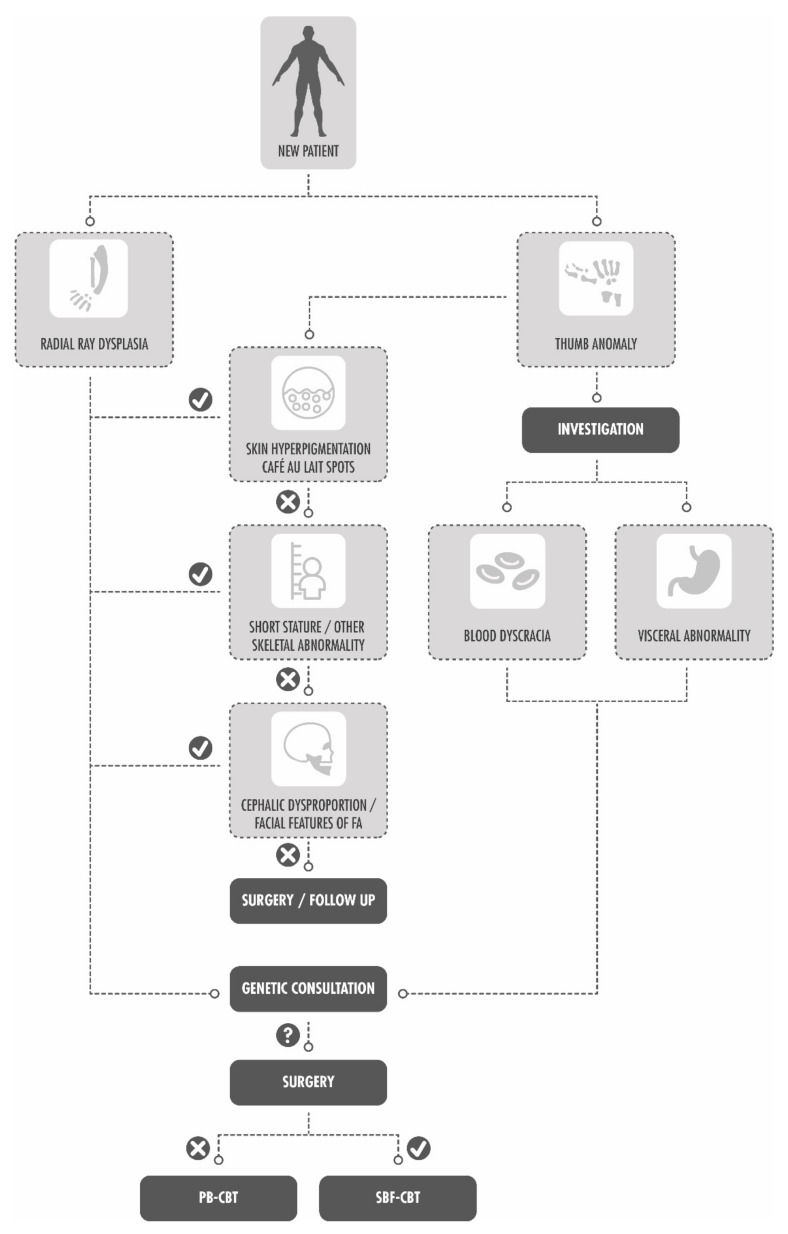
Algorithm for Fanconi anemia testing of patients with radial ray dysplasia or thumb anomalies presenting to the Congenital Hand Department at our hospital. PB-CBT = Peripheral Blood Chromosome Breakage Test. SBF-CBT = Skin Biopsy Fibroblast Chromosome Breakage Test.

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
