# Peer review of "Fanconi Anemia: Examining Guidelines for Testing All Patients with Hand Anomalies Using a Machine Learning Approach"

_children, 2022, doi:10.3390/children9010085_

Round 1
Reviewer 1 Report
Line 82 have to clarified
Author Response
Dear Ms. Zhang, dear Reviewers,
we are pleased to resubmit our paper entitled “Fanconi Anemia: Examining Guidelines for Testing all Patients with Hand Anomalies Using a Machine Learning Approach” to Children with changes according to the reviewer’s suggestions.
We would like to thank the reviewers and appreciate their invested time and comments for our manuscript. We will now respond to the reviewers’ comments point-by-point:
Comments of the Reviewers:
Reviewer #1:
Line 82 have to clarified
We thank the reviewer for the valuable input and improved statement in line 82. We wanted to transport the information that only 4 patients referred to our unit had just hand anomalies and no other features associated with FA.
Reviewer #2:
The hand and skin anomalies are those more easily identificable, and in the past were maybe the first sign of FA. Visceral abnormalities, short stature, cephalic dysproportion and blood dyscracia are nowdays identified early?
We thank for this comment and agree that this might be one reason. It underlines our statement that hand anomalies should not be interpreted as a predictor for FA.
In the article text I suggest to add:
In introduction incidence/year/100.000 or prevalence
We thank the reviewer for this input and added the incidence into the introduction. The incidence of FA is estimated to be around 1 in 136,000 births.
In methods population served by the hospital, or location.
In the original manuscript we added the hospital name what is not allowed by the editor to keep the manuscript anonymously in the review process. However, a statement about the size of the hospital is added to the materials section (paragraph 2.1).
In methods/results the date of begin and end of the study
We apologize for the confusion we have caused. We added a statement about the begin and end of collection dates of the patients to the methods section.
In methods/results the advantages and problems of using a "machine learning approach"
Dear reviewer, we are happy that we can further discuss the main principle of this study. We now further evaluated this method for the study in the third paragraph of the discussion section, where an evaluation of this method is placed the best.
We thank you for your invested time and considering our work for your journal.
Sincerely yours

Reviewer 2 Report
This study bring to costructive comments on guidelines about Fanconi Anemia, the conclusions in flow chart can bring to save in cost of exams and stress for patients and families, reducing the number of test for positive or negative patients. Doctors as well can benefit from an agreement on protocols. I appreciate reading.
The hand and skin anomalies are those more easily identificable, and in the past were maybe the first sign of FA. Visceral abnormalities, short stature, cephalic dysproportion and blood dyscracia are nowdays identified early?
In the article text I suggest to add:
In introduction incidence/year/100.000 or prevalence
In methods population served by the hospital, or location.
In methods/results the date of begin and end of the study
In methods/results the advantages and problems of using a "machine learning approach"
Author Response

(The authors gave the same response as above.)
